# Truth Telling about Tobacco and Nicotine

**DOI:** 10.3390/ijerph16040530

**Published:** 2019-02-13

**Authors:** Rachelle Annechino, Tamar M.J. Antin

**Affiliations:** 1Critical Public Health Research Group, Prevention Research Center, 2150 Shattuck Avenue, Suite 601, Berkeley, CA 94704, USA; 2Center for Critical Public Health, Institute for Scientific Analysis, 1150 Ballena Boulevard, Suite 211, Alameda, CA 94501, USA; tamar@criticalpublichealth.org

**Keywords:** tobacco, nicotine, e-cigarettes, ENDS, harm reduction, trust

## Abstract

Research suggests that many people in the US are misinformed about the relative harms of various tobacco and nicotine products. Concerns about public misinformation have often been framed as relevant only to the degree that public health institutions agree to prioritize conventional approaches to tobacco harm reduction. We argue that while the information priorities of public health professionals are important, ethical and credible information sharing also requires consideration of broader issues related to public trust. To promote trust, public health institutions must develop truth telling relationships with the communities they serve and be genuinely responsive to what people themselves want to know about tobacco and nicotine products.

Debate over public health approaches to tobacco and nicotine regulation has intensified over the past decade with the rise of commercial vaping in the form of electronic cigarettes or “e-cigarettes” (which we use here to refer collectively to systems designed for the delivery of aerosols that may or may not contain nicotine, such as cigalikes, tanks and mods) and tobacco heat-not-burn products. The range of public health perspectives on tobacco and nicotine may be interpreted in multiple ways, but generally a position favoring regulation conditioned on relative harms, often characterized as “harm reduction”, is contrasted with a position that favors more prohibitive, relatively uniform regulation of all such products. Conflict between these positions has surfaced in public health discussions internationally via the World Health Organization’s Framework Convention on Tobacco Control, and nationally in countries including Australia, India and the European Union [1,2,3,4]. In the US, the 2018 Annual Review of Public Health exemplifies prominent perspectives in this debate with its presentation of Glantz & Bareham’s [5] more prohibitive “precautionary principle” viewpoint alongside Abrams et al.’s [6] contrasting “harm minimization” or harm reduction viewpoint. As Green et al.’s discussion suggests, major policy differences in this dispute may be traced to different risk management philosophies [7].

In one area, however, the contours of debate are less clear. Abrams et al., for example, give prominence to the need for “accurate public information” [6], a theme that is echoed in other harm reduction-oriented publications with repeated calls for more “candid” public communication, and condemnation of “misleading risk communications” [8,9,10,11]. In line with this theme, evidence suggests that many people, including tobacco control professionals, misperceive the relative health risks of non-combustible products such as e-cigarettes, snus and nicotine replacement therapies, particularly as compared to combustibles, and over attribute the health harms of tobacco and nicotine products to the presence of nicotine alone [9,12,13,14,15,16]. In some settings, the prevalence of inaccurate and conflicting beliefs about the harms of some nicotine and non-combustible tobacco products has also increased in recent years, suggesting that some people may have been misinformed more often than they have been informed about these products, whether through public health, mass media, and/or social communication [15,17,18]. 

Regardless of one’s stance on harm reduction, it should go without saying that public health representatives are committed to sharing accurate information. Despite disagreements over public health approaches to tobacco control, there is widespread agreement among researchers on the existence of significant differences in the harmfulness of different tobacco and nicotine products. Influential figures on all sides of the e-cigarette debate in the US, for example, have increasingly acknowledged a body of evidence that vaping can be less harmful than combustible cigarettes [19]. Nevertheless, the tobacco control community remains divided over questions about how to communicate, or even whether to communicate, information about the relative risks of tobacco and nicotine products [20]. In our own research [21], people who vape have expressed dismay at public health campaigns:


*“They could just say [vaping is] less harmful, right? … But then they downplay it again… And that’s like, we all know that. But it’s still healthier than the latter, than the cigarettes. So, just let me make my decision is what I’m saying. Give me all of the facts, all of the facts, and I’ll make a decision.”*
(from an interview for the “Electronic Nicotine Delivery Systems and California Youth” study [21])

Of course, individuals with different orientations to tobacco and nicotine policies may have different takes on which facts are relevant, which may produce different interpretations of what constitutes accurate or truthful information. From a precautionary principle viewpoint [5,7], understanding nicotine in all its various forms of consumption as addictive or potentially toxic may be considered a truth singularly relevant to public communication, while information regarding differential product harms may be considered irrelevant. From a harm minimization viewpoint [6,7], meanwhile, information about differential product harms is more relevant. It is unfortunate, however, that concerns about public communication of information have often been expressed exclusively in terms of this debate within public health, while the perspectives of the communities served by public health have received less attention. In this top down framing, concerns that people are misinformed regarding differential harms may appear relevant only to the degree that public health institutions agree to support tobacco harm reduction. What people themselves want to know about such products, meanwhile, may appear to be of little import.

Yet public health communication cannot be guided solely by the priorities of public health professionals. When institutions do not also listen to and engage with the interests of people with different world views who are served by public health, they risk communicating in ways that are not only unethical, but ineffective [22,23,24]. Furthermore, loss of credibility has its own consequences that may not be adequately captured in models of aggregate population effects, nor even confined to the realm of tobacco-related diseases. The relationship between effectiveness and trust is highlighted in the American Public Health Association’s “Principles of the Ethical Practice of Public Health” as it describes how institutions charged with protecting public health can communicate in ways that either nurture public trust or inhibit it:


*“The effectiveness of institutions depends heavily on the public’s trust. Factors that contribute to trust in an institution include the following actions on the part of the institution: communication; truth telling; transparency (i.e., not concealing information); accountability; reliability; and reciprocity. One critical form of reciprocity and communication is listening to as well as speaking with the community.”*
[25]

Notably, the points above are about actions, and the relationships institutions build with the people they serve. Following this model, the trustworthiness of an institution need not depend upon a particular conception of the truth (which may vary with one’s information priorities), so much as it relies on reciprocal communication or truth telling. In biomedical ethics, the concept of “truth telling” is often used to describe an active process for informing patients about a terminal health condition. For example, where the paternalistic withholding of information from patients with terminal conditions was once accepted in the US, medical practitioners are now expected to tell patients what patients want to know, even if practitioners believe their patients would be better off without such information [26]. Similarly, sharing information about nicotine and tobacco products based solely upon what one presumes to be a community’s “own good” is not truth-telling; to tell someone the truth is to be responsive to the interests of one’s audience. Perceptions of an institution’s motivations and trustworthiness are damaged when it promotes messages that reinforce rather than correct misconceptions, or when it otherwise obscures, neglects or misrepresents information that people consider relevant to their own interests [27,28,29]. In public health research too, reports on findings that do not account for a community’s concerns can also impede trust [21].

In recent decades confidence in public health communication about tobacco and nicotine has been bolstered by high levels of distrust in the tobacco industry. At the same time, public health institutions must consider their own histories in developing trust relationships with the communities they serve. In the US for example, health institutions must account for damaged credibility and distrust sown through a long history of deception and exploitation of marginalized people, with consequences impacting groups that are also disproportionately impacted by tobacco-related illness [30,31]. Insofar as people affected by smoking disparities want accurate information about the differential harms of tobacco and nicotine products, the eliding of these distinctions by public health representatives may further erode trust. Similar to the historical record of medical mistreatment and research enacted upon disadvantaged groups without their informed consent, when people are misinformed on topics that they themselves regard as relevant, they cannot make informed choices—not only in their individual lives, but also as participants in community policymaking. Moreover, trust issues in public health occur against a backdrop of increased access to powerful tools for spreading disinformation and eroding public trust [32,33,34,35]. In an era in which community “gate watchers” and “social proof” are at least as important as traditional gatekeepers or credentialing as arbiters of expertise, signals of trustworthiness such as transparency and a willingness to seek out and engage with diverse points of view may become even more salient [36,37]. In this setting, a commitment to truth telling that is responsive to public interests is especially critical.

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
