# Peer review of "Truth Telling about Tobacco and Nicotine"

_ijerph, 2019, doi:10.3390/ijerph16040530_

Reviewer 1 Report

I enjoyed reading this article of a very hot and topical subject.

I felt the authors gave a balanced viewpoint of the different public health perspectives whilst coming down strongly in favour of presenting information to the public that they want and need in order to make their own informed choices, rather than the professionals pursuing their own agendas. I found the comparison with biomedical ethics, and how that has changed from a paternalistic approach, to that where patients are told what they want to know particularly pertinent.

I have made a few comments below. The authors may wish to discuss some of these unless they consider these to be outside the scope of their article. In particular, as I mention below, I think it might benefit from a wider perspective than just the US.

Although this article has a clear US perspective, and the debate around e-cigarettes does seem particularly diverse in the US, this is of global relevance, with similar debates seen elsewhere, for example in Australia. Therefore I wondered if the authors would consider mentioning that this debate isn’t confined to the US, and give some examples from elsewhere.

In addition, even where the public health community seem to be more in agreement and in the messages they portray, there can still many misconceptions around relative harms in the general population. For example in the UK with Public Health England evidence review https://www.gov.uk/government/publications/e-cigarettes-and-heated-tobacco-products-evidence-review), and even a UK government select committee report that was informed by many expert and lay opinions https://www.parliament.uk/business/committees/committees-a-z/commons-select/science-and-technology-committee/news-parliament-2017/e-cigarettes-report-publication-17-19/), a large proportion of the public still think that e-cigarettes are as harmful as smoking and that nicotine is the most harmful constituent of cigarettes. So clearly even in countries where public health are giving a strong message out, this doesn’t always reach the public, or perhaps they choose to take other viewpoints into consideration (for example conflicting messages from the media and social media).

Finally, researchers also have an obligation to present the 'truth' and are not immune from this phenomenon. Published e-cigarette research (particularly animal research) has sometimes drawn conclusions way beyond that which is reasonable. If this is then picked up by the media incorrect messages may be magnified even further, resulting in further public confusion.

Author Response

The authors wish to thank the reviewers for their very helpful comments. We have made the following changes in response to their feedback.

Reviewer 1

Although this article has a clear US perspective, and the debate around e-cigarettes does seem particularly diverse in the US, this is of global relevance, with similar debates seen elsewhere, for example in Australia. Therefore I wondered if the authors would consider mentioning that this debate isn’t confined to the US, and give some examples from elsewhere.

We expanded the frame of reference at several points in the article (lines 26-29, lines 37-45, line 104, lines 114-119) that were previously limited to the US.

In addition, even where the public health community seem to be more in agreement and in the messages they portray, there can still many misconceptions around relative harms in the general population. For example in the UK with Public Health England evidence review https://www.gov.uk/government/publications/e-cigarettes-and-heated-tobacco-productsevidence-review), and even a UK government select committee report that was informed by many expert and lay opinions https://www.parliament.uk/business/committees/committeesa-z/commons-select/science-and-technology-committee/news-parliament-2017/e-cigarettesreport-publication-17-19/), a large proportion of the public still think that e-cigarettes are as harmful as smoking and that nicotine is the most harmful constituent of cigarettes. So clearly even in countries where public health are giving a strong message out, this doesn’t always reach the public, or perhaps they choose to take other viewpoints into consideration (for example conflicting messages from the media and social media).

We included these and other references to public misconceptions outside the US, and provided additional context on why these misconceptions may exist (lines 37-45).

Finally, researchers also have an obligation to present the 'truth' and are not immune from this phenomenon. Published e-cigarette research (particularly animal research) has sometimes drawn conclusions way beyond that which is reasonable. If this is then picked up by the media incorrect messages may be magnified even further, resulting in further public confusion.

We added an additional sentence to draw attention to this important point (lines 101-102).

Reviewer 2 Report

This is a very well written commentary on a real ‘hot topic’ in public health globally at this moment in time. The authors capture well the two sides of the dualistic subject positions that public health professionals have taken towards alternative nicotine delivery devices. I thought that the objectivity of the commentary was commendable and that the authors stance that public health institutions should respond to the health information needs of the public they serve was an extremely useful and balanced position to take. The authors may be interested to read a recent commentary published in the Harm Reduction Journal that suggests that peer involvement in research and service development is critical, as this draws on a similar stance: https://harmreductionjournal.biomedcentral.com/articles/10.1186/s12954-018-0275-1 

I have only minor comments and suggestions as otherwise think this is an excellent commentary that I would like to see published.

Reading this as an academic working in tobacco harm reduction I am familiar with the debates and positions outlined. The only aspect that jarred was the idea of ‘truth telling’. I understand that within an ethical / moral framework this term has specific meaning, but not being aware of this, or, for me, having to dig deeper into the references to understand how this is defined and understood, was not clear enough. Therefore my preference would be to use different terminology here, perhaps ‘communicating the evidence objectively’, or ‘transparent reporting of research findings’ might be more appropriate? If the authors decide to stick with the term ‘truth-telling’ this should be clearly defined up front, otherwise I was distracted by thinking what is the truth? and how can we be sure that what we are telling is ‘the truth’?. Is there more than one truth? There are too many debatable positions for the term to be useful without clear definition

Is it correct to call communities ‘publics’? This term doesn’t seem quite right.

There is convincing evidence from the UK, as well as the US, that public perceptions of the harms of nicotine are inaccurate. The authors may wish to include a reference to this evidence to make the commentary more applicable to an international audience. https://assets.publishing.service.gov.uk/government/uploads/system/uploads/attachment_data/file/684963/Evidence_review_of_e-cigarettes_and_heated_tobacco_products_2018.pdf

Minor typos:

Pg 1 Line 26. Full stop needed after ‘products’.

Page 2. Quotation needs acknowledgement and context. What study is this drawn from?

Pg 3, line 4. Duplication of the word ‘that.

Author Response

The authors wish to thank the reviewers for their very helpful comments. We have made the following changes in response to their feedback.

Reviewer 2

Reading this as an academic working in tobacco harm reduction I am familiar with the debates and positions outlined. The only aspect that jarred was the idea of ‘truth telling’. I understand that within an ethical / moral framework this term has specific meaning, but not being aware of this, or, for me, having to dig deeper into the references to understand how this is defined and understood, was not clear enough. Therefore my preference would be to use different terminology here, perhaps ‘communicating the evidence objectively’, or ‘transparent reporting of research findings’ might be more appropriate? If the authors decide to stick with the term ‘truth-telling’ this should be clearly defined up front, otherwise I was distracted by thinking what is the truth? and how can we be sure that what we are telling is ‘the truth’?. Is there more than one truth? There are too many debatable positions for the term to be useful without clear definition.

We appreciate this insight and have now expanded on this concept of truth telling to improve clarity in lines 90-98.

Is it correct to call communities ‘publics’? This term doesn’t seem quite right.

We changed "publics" to "communities" (line 15, line 69).

There is convincing evidence from the UK, as well as the US, that public perceptions of the harms of nicotine are inaccurate. The authors may wish to include a reference to this evidence to make the commentary more applicable to an international audience. https://assets.publishing.service.gov.uk/government/uploads/system/uploads/attachment_data/file/684963/Evidence_review_of_ecigarettes_and_heated_tobacco_products_2018.pdf

We added this reference and changed the language to include areas outside the US (lines 37-45).

Minor typos:
Pg 1 Line 26. Full stop needed after ‘products’.
Page 2. Quotation needs acknowledgement and context. What study is this drawn from?
Pg 3, line 4. Duplication of the word ‘that.

We made all of these changes.